# Effects of Fat Pre-Emulsification on the Growth Performance, Serum Biochemical Index, Digestive Enzyme Activities, Nutrient Utilization, and Standardized Ileal Digestibility of Amino Acids in Pekin Ducks Fed Diets with Different Fat Sources

**DOI:** 10.3390/ani12202729

**Published:** 2022-10-11

**Authors:** Xiangyi Zeng, Keying Zhang, Gang Tian, Xuemei Ding, Shiping Bai, Jianping Wang, Li Lv, Yupeng Liao, Yue Xuan, Qiufeng Zeng

**Affiliations:** 1Institute of Animal Nutrition, Key Laboratory for Animal Disease-Resistance Nutrition of China Ministry of Education, Sichuan Agricultural University, 211 Huimin Road, Wenjiang District, Chengdu 611130, China; 2Si Chuan Action Biotech Co., Ltd., 11 Jinxing Road, Guanghan City, Deyang 618302, China

**Keywords:** ducks, fat pre-emulsification, duck fat, growth performance, nutrient utilization

## Abstract

**Simple Summary:**

Emulsifiers, which can reduce fat chylomicrons and improve fat emulsification, are currently widely used in animal feed to increase the utilization of fats and improve animal growth performance. However, the traditional method of adding emulsifiers directly to feed has shown inconsistent effects on livestock and poultry. Moreover, China is the largest duck-producing country, and it can, therefore, be anticipated that many duck processing enterprises will produce a large amount of duck fat. However, there is no information about the effects of duck fat as a fat source for Pekin duck. Considering the increase in oil prices, this study was conducted to investigate the effects of fat pre-emulsification (preE), which is a new method to improve the utilization of dietary oil, on Pekin ducks fed diets with different fat sources. The obtained results revealed that fat preE contributed to the utilization of dietary nutrients, serum lipid metabolism, intestinal digestive enzyme activities, and liver health, thereby improving the growth performance of ducks; duck fat has higher bioavailability for ducks based on dietary ether extract (EE) utilization.

**Abstract:**

This experiment was conducted to evaluate the effects of fat pre-emulsification on growth performance, the serum biochemical index, intestinal digestive enzyme activities, nutrient utilization, and the standardized ileal digestibility of amino acids (SIDAA) in Pekin ducks fed diets containing different fat sources. Three hundred and twenty healthy ten-day-old Pekin male ducks (409 ± 27 g) were assigned to a 2 × 2 factorial design and given one of two types of poultry fat (duck fat or a mixed type of fat composed of chicken fat and duck fat in a 1:1 ratio) that had been pre-emulsified or not. This resulted in four treatments of eight replicates, with each replicate having ten ducks. The results showed that fat pre-emulsification (preE) significantly increased (*p* < 0.05) body weight and body weight gain and decreased (*p* < 0.05) the feed-to-gain ratio, the liver index, the activity of aspartate aminotransferase (AST) and the concentration of total cholesterol (TC) in the serum. Fat preE also tended to decrease the activity of lipase (*p* = 0.07) and significantly reduced (*p < 0.05*) the activity of trypsin in the duodenum. The utilization of dietary dry matter, ether extract (EE), energy, and total phosphorus, as well as apparent metabolizable energy (AME) and the SID of serine (*p* = 0.090), were improved by fat preE. Duck fat markedly increased (*p* < 0.05) the serum TC concentration and the utilization of dietary EE; however, it also tended to decrease the serum triglyceride (TG) concentration (*p* = 0.09) and markedly decreased (*p* < 0.05) the activity of trypsin in the jejunum and duodenum. These results indicated that fat preE contributed to the utilization of dietary nutrients, serum lipid metabolism, intestinal digestive enzyme activities, and liver health, thereby improving the growth performance of ducks. Duck fat has higher bioavailability for ducks based on dietary EE utilization.

## 1. Introduction

Lipids (fats and oils) are the main energy source for poultry, with an energy value that is at least twice as high as those of carbohydrates and protein [1]. The lipids added to poultry diets can increase growth rates and feed conversion efficiency [2,3]. With the increase in the cost of raw feed materials in recent years, the proportions of unconventional raw materials in feed have increased, and more amounts of fat have had to be added to meet the requirements in metabolizable energy (ME) for fast-growing birds and improve feed conversion efficiency because of the limited formulation space [4,5]. However, the excessive addition of fat in the diet can result in the failure of the animals to make full use of dietary lipids, which not only leads to fat waste and increased breeding costs but can also cause diarrhea in livestock and poultry [6,7].

Emulsifiers, which can reduce the surface tension of fat and water due to their physicochemical properties of being both hydrophilic and lipophilic, facilitate the formation of micelles and improve the digestibility and absorption of fat in livestock and poultry [8]. Kaczmarek et al. [9] reported that the diets supplemented with glyceryl polyethylene glycol ricinoleate improved the production performance and resulted in higher apparent total tract digestibility (ATTD) of crude fat. However, some reports on livestock and poultry showed that exogenous emulsifiers had no significant effect on productive indicators [10,11,12,13]. It was found that the effects of emulsifiers on livestock and poultry varied greatly and were affected by many factors (e.g., the type of emulsifier, the supplemented dose of emulsifier, and the age and strain of poultry) [14].

Additionally, Mun et al. [15] suggested that the conventional method of adding emulsifiers for lipid emulsification in the gastrointestinal tract is difficult to determine and control, which plays an important role in determining the rate and extent of lipid hydrolysis. More interestingly, in the food industry, Garaiova et al. [16] reported that pre-emulsification (preE) of an oil mixture prior to ingestion may enhance the digestion and absorption of longer chain, more highly unsaturated fatty acids in healthy adults, suggesting that preE of fish oils may be a useful means of boosting the absorption of fatty acids. Thus, the preE of fats or oils is a new emulsification method. A mixture of fat or oil, an exogenous emulsifier, and water in a certain proportion is rapidly stirred with a homogenizer for conversion into emulsified fat or oil, which is then added to livestock and poultry feed. However, to the best of our knowledge, no studies have investigated the effect of fat or oil preE on livestock and poultry.

Zhang et al. [17] reported that the birds fed diets supplemented with poultry fat had similar growth performance to those fed diets supplemented with soybean oil, suggesting that poultry fat was a good fat source for chickens. Therefore, because of the recent increase in the price of lipids, there is a greater interest in adding poultry fat instead of soybean oil to the diet as Chinese duck nutritionists strive to reduce breeding costs. Moreover, China is the largest duck-producing country, accounting for more than 75% of the global duck market [18]. It can, therefore, be anticipated that many duck-processing enterprises will produce a large amount of duck fat. If it could be used in feed, duck fat would fill the gap in energy feed in China to a certain extent. However, there is no information about the effects of duck fat as a fat source for Pekin duck. Currently, it is widely believed that duck fat and chicken fat have similar nutritional values for ducks, and duck fat is often mixed with chicken fat in a large amount of small oil refining enterprises in China. However, different sources of fat have different compositions of fatty acids (FAs), which can affect the growth performance and digestibility of fat [19,20], as well as the effects of emulsifiers on poultry [14]. Guerreiro Neto et al. [21] also reported that emulsifiers are more effective in bird diets supplemented with soybean oil (rich in polyunsaturated FAs) than in those formulated with palm oil (rich in saturated FAs). The saturated FAs in dietary lipids are relatively slowly incorporated into micelles, compared with polyunsaturated FAs, due to their property of being nonpolar, which increases their requirement for bile salts for effective emulsification [22,23]. Diets formulated with different fat sources and exogenous emulsifiers have aroused intense interest regarding their contribution to improving lipid metabolism and performance [5,18,24]. Therefore, the objective of the present study was to evaluate the effects of fat preE on Pekin ducks fed diets containing different fat sources on their growth performance, serum biochemical index, intestinal digestive enzyme activities, nutrient utilization, and standardized ileal digestibility of amino acids (SIDAA).

## 2. Materials and Methods

The study was approved by the Animal Care and Use Committee, Sichuan Agricultural University (Ethic Approval Code: SICAUAC202110-2; Chengdu, China).

### 2.1. Birds, Diets, and Management

One-day-old male Pekin ducks were fed a standard starter diet containing 11.93 MJ/kg ME and 19.50% CP from 1 to 10 days of age. On Day 10, 320 healthy ten-day-old Pekin male ducks (409 ± 27 g) were randomly assigned to a 2 × 2 factorial design and fed 1 of 2 types of poultry fat (duck fat vs. a mixed type of poultry fat composed of chicken fat and duck fat in a 1:1 ratio) that was pre-emulsified or not. This resulted in 4 treatments of 8 replicate cages with 10 ducks per cage. Pre-emulsified fat: fat, an emulsifier, and water were added at a ratio of 150:3:25, after which the mixture was stirred with a homogenizer at 3000 r/min for 20 s. The emulsifier was provided by Si Chuan Action Biotech Co., Ltd. and contained 41% propionic acid, 24% ammonium propionate, and 10% polyethylene glycol glycerine ricinoleate.

The experimental diets were formulated according to the NRC (1994) and provided in pellet form. The composition and nutrient contents of the experimental diets are shown in Table 1. The fatty acid profiles of duck fat and mixed fat, which were determined using gas chromatography as described previously by Yang et al. [25], are shown in Table 2. All the ducks were reared in cages (1.0 × 0.8 × 0.6 m) with a “23 h on–1 h off” lighting regimen for the first 3 d and then under a “16 h on–8 h off” lighting regimen for the remainder of the feeding period in a temperature- and humidity-controlled room throughout the experiment and provided feed and water ad libitum during the whole experimental period.

### 2.2. Sample Collection and Determination

At 34 days of age, after fasting for 12 h, all the ducks were weighed, and feed consumption was determined on a cage basis for the calculation of the body weight (BW), body weight gain (BWG), feed intake (FI), and feed-to-gain ratio (F/G). The weights of the birds that died during the experiment were recorded, and the data were used to adjust the F/G.

Then, eight ducklings per treatment (one bird per cage) with body weights close to the average of each cage were selected for blood sampling via the jugular vein. The serum was obtained via centrifugation at 3000 r/min at 4 °C for 10 min, and then serum samples were stored at −20 °C until the determination of the serum biochemical index. The activities of alanine aminotransferase (ALT) and aspartate aminotransferase (AST) and the contents of total cholesterol (TC), triglyceride (TG), high-density lipoprotein (HDL), low-density lipoprotein (LDL), very low-density lipoprotein (VLDL), and total bile acid (TBA) were analyzed using an automatic biochemical analyzer (HATICHI 7180, Tokyo, Japan).

The ducks from which blood was collected were euthanized through exsanguination. Liver weights were obtained and calculated based on live BW. After that, one duckling per cage (a total of eight per treatment) was randomly chosen and bled for collecting the digesta from the duodenum and jejunum, and the digesta samples were stored at −80 °C for enzyme activity analysis. The activities of trypsin and lipase in the duodenum and jejunum were determined according to the instructions provided with the obtained commercial assay kits (Nanjing Jiancheng Bioengineering Institute, Nanjing, China).

### 2.3. Assays to Determine the Standardized Ileal Amino Acid Digestibility and Nutrient Utilization of the Diets

On Day 35, two ducks randomly selected from each replicate were reared in metabolic cages (two ducks per cage) and fed the original diets mixed with 0.5% TiO_2_ as an indigestible marker to determine nutrient utilization. After acclimation for 2 d, the excreta were collected on a cage basis for 72 h. After removing debris such as feathers and feed, all the excreta samples were mixed according to the cage and immediately put into the refrigerator at −20 °C for preservation. Four experimental diets and all the excreta samples were analyzed for TiO_2_ contents according to the method from Short et al. [26]. Then, all the diets and excreta samples were analyzed for dry matter (DM) (method 930.15) [27], N (method 968.06) [28], EE (method 934.01) [28], Ca (method 984.01) [29], and P (method 965.17) [28]. The nitrogen content analysis of all the samples was performed using a machine (Kjeltec 2300 Nitrogen Analyzer; Foss Tecator AB, Hoeganaes, Sweden). Crude protein was calculated as N × 6.25. Ether extract (EE) in diets and excreta was measured with a Soxhlet apparatus for approximately 8 h. All the samples were also analyzed for gross energy (GE) using Parr 6400 oxygen bomb calorimeter (Parr Instrument Co., Moline, IL, USA). The nutrient utilization of all diets was calculated using the following formula: nutrient utilization (%) = {1 − [(N_e_ × T_d_)/(N_d_ × T_e_)]} × 100, where T_e_ = TiO_2_ contents in the excreta (% DM), T_d_ = TiO_2_ contents in the diet (% DM), N_e_ = nutrient concentration in the excreta (% DM), and N_d_ = nutrient concentration in the diet (% DM). The AME of the experimental diets were calculated using the analyzed content of TiO_2_ and GE as follows: AME = GE_d_ − [(GE_e_) × (T_d_/T_e_)], where = GE_d_ is the GE in the diet (% DM), GEe is the GE (kcal/kg) in the excreta (% DM).On Day 40, after the excreta samples were collected, all the ducks were used to perform the digestibility trial, and 16 healthy ducks (8 replicates; 2 ducks per replicate) were fed a free-nitrogen diet mixed with TiO_2_ (0.5%) after fasting for 8 h; they were fed for 4 h and then euthanized via cervical dislocation. The ileal digesta was gently rinsed with distilled water into plastic containers [30]. The collected ileal samples from 2 birds within a cage were pooled and stored at −20 °C for the subsequent analyses of DM, TiO_2_, and amino acids (AAs). For AA (Lys: lysine; Met: methionine; Arg: arginine; Ile: isoleucine; Leu: leucine; Thr: threonine; Val: valine; Phe: phenylalanine; His: histidine; Asp: aspartic acid; Ser: serine; Glu: glutamic acid; Gly: glycine; Ala: alanine; Cys: cysteine; Tyr: tyrosine; Pro: proline) analyses, the diets and ileal digesta were hydrolyzed with 6 N HCl for 24 h at 110 °C (method 982.30 E; AOAC International, 2005) and filtered, and the AA contents were analyzed with an automatic amino acid analyzer (HITACHI L-8900) according to Zhang et al. [31]. The TiO_2_ contents of all the diets and ileal digesta samples were determined according to the method from Short et al. [26]. These data were used to calculate the SIDAA based on our previous studies (Han et al. [32] and Qin et al. [30]). The apparent and standardized ileal digestibility (AID and SID, %) and basal ileal endogenous losses (BELs) of the AA in the assay diets were calculated according to the following equations: AID = [1 − (A_I_ × I_D_)/(I_I_ × A_D_)] × 100; BEL = A_I_ × (I_D_/I_I_); SID = AID + [100 × (BEL/A_D_)], where I_D_ is the content of TiO_2_ in the diet (g/kg DM); A_I_ is the concentration of AA in the ileal digesta (g/kg DM); I_I_ is the content of TiO_2_ in the ileal digesta (g/kg DM); and A_D_ is the concentration of AA in the diet (g/kg DM).

### 2.4. Statistical Analysis

Data were analyzed with two-way ANOVA using the GLM procedure of SAS (SAS Institute Inc., Cary, NC, USA). Each cage was considered an experimental unit. The models included the main effects of the fat sources and fat preE, as well as two-way interactions between fat source and fat preE. The data are expressed as the mean ± SEM. The probability of *p* < 0.05 was described as significant, and 0.05 < *p* < 0.1 was described as a trend.

## 3. Results

### 3.1. Growth Performance

The effects of fat sources and fat preE on BW, BWG, FI, and F/G are displayed in Table 3. Duck fat had no significant effect (*p* > 0.05) on the growth performance of the ducks from 11 to 34 days of age. Fat preE significantly increased (*p* < 0.05) BW at 34 days of age as well as the BWG of the ducks from 11 to 34 days of age and decreased (*p* < 0.05) the F/G. There was no interaction (*p* > 0.05) between fat source and fat preE on the growth performance of the ducks.

### 3.2. Liver Index and Serum Biochemical Parameters

Fat preE significantly decreased (*p* < 0.05) the liver index and the activity of AST but did not influence serum ALT activity (Figure 1). As shown in Table 4, the ducks fed diets containing duck fat had higher TC concentrations (*p* < 0.05) than those fed diets containing mixed poultry fat. Fat preE decreased (*p* < 0.05) the content of TC in the serum, and duck fat showed a tendency (*p* = 0.09) to decrease the TG concentration. The activity of ALT and the concentrations of HDL-C, LDL-C, VLDL-C as well as the level of TBAs in the serum were not influenced (*p* > 0.05) by either fat source or fat preE. Additionally, a significant interaction (*p* < 0.05) between fat source and fat preE was observed for the serum TC concentration, in which fat preE decreased the TC concentration in the ducks fed diets with duck fat rather than diets with mixed fat.

### 3.3. Intestinal Digestive Enzyme Activities

The intestinal digestive enzyme activity results are presented in Table 5. The duck fat diets decreased (*p* < 0.05) the activity of trypsin in the duodenum and jejunum. However, the activity of lipase in the duodenum and jejunum was not influenced (*p* > 0.05) by fat source. Fat preE reduced (*p* < 0.05) the activity of trypsin in the duodenum and tended to decrease (*p* = 0.07) the activity of lipase in the duodenum. No interaction (*p* > 0.05) was observed between fat source and fat preE on the intestinal digestive enzyme activities.

### 3.4. Nutrient Utilization

The diets supplemented with duck fat increased (*p* < 0.05) the utilization of EE; however, they had no significant effect (*p* > 0.05; Table 6) on the utilization of DM, TP, or AME. Fat preE improved (*p* < 0.05) the utilization of DM, EE, TP, and AME. Ca and CP utilization rates were not affected (*p* > 0.05) by either fat source or preE. No interaction effect was observed (*p* > 0.05) between fat source and fat preE on the utilization of nutrients and AME.

### 3.5. Standardized Ileal Digestibility of Amino Acids

Decreases in the SID of Asp, Glu, Tyr, Val, Lys, Ile, Phe, His, the total essential AAs (EAAs), and the total AAs were observed (*p* < 0.05; Table 7; Table 8) in the duck fat diet groups, and duck fat tended to decrease the SID of Leu and total nonessential AAs (NEAAs); however, the ducks fed diets formulated with duck fat had higher SID of Cys than those fed diets formulated with mixed fat (*p* < 0.05). Fat preE tended to increase (*p =* 0.09) the SID of Ser, whereas it had no significant influence (*p* > 0.05) on the SID of other AAs. An interaction effect (*p* > 0.05) between fat source and preE was not observed on the SIDAA in ducks.

## 4. Discussion

In the current study, fat preE significantly increased the BW and BWG and decreased the F/G of the ducks from 11 to 34 days of age but did not affect the FI. Similar results were observed by Hu et al. [24], who found that adding an emulsifier increased the BW and BWG and decreased the F/G of 18- to 42-day-old meat ducks, but the ADFI was not influenced. Kaczmarek et al. [9] found that the birds fed diets supplemented with glyceryl polyethylene glycol ricinoleate (GPR) exhibited higher BWG and lower FCR than the chickens receiving diets without GPR during the grower period (d 14–35). Haetinger et al. [5] demonstrated that synthetic emulsifier supplementation could enhance the performance of broilers by increasing the energy values of diets and the digestibility of DM, protein, and fat. Similarly, in the present study, fat preE enhanced the AME of the diets and improved the utilization of DM and EE, which was consistent in part with findings by Wealleans et al. [33], who observed increases in AME, DM, N, and fat digestibility when broilers were fed diets formulated with bioemulsifier on Day 21.

Calcium and phosphorus account for more than 70% of the total ash in the body and play important roles in the growth development of poultry [34]. In this experiment, the utilization of dietary TP was higher for the ducks receiving diets with fat preE in comparison with the basal diet. The reason may be related to the improvement in intestinal health. Many authors have observed that the birds receiving diets formulated with exogenous emulsifiers exhibit better intestinal morphology and antioxidant capacity [35], along with decreased numbers of harmful bacteria in the intestinal tract [36] and improved intestinal mucosal absorption areas [37], thus indirectly or directly promoting the digestibility and absorption of nutrients and improving growth performance.

Lai et al. [38] reported that the activities of duodenum lipase and lipoprotein lipase were increased by the dietary levels of bile acids during the starter and growth phases, which suggested that the intestinal lipase activity can be an indicator of lipid utilization in animals. Inconsistently, we found that fat preE tended to decrease the activity of lipase and trypsin in the duodenum in the present study. The reason may be that fat preE increased the rates of the degradation and digestibility of dietary lipids, thus saving intestinal digestive enzyme synthesis and secretion and resulting in decreased lipase activity. Lipase is a special ester hydrolase that acts on the ester bonds of TGs, degrading them into diglycerides, monoglycerides, glycerol, and fatty acids and controlling digestion, absorption, fat remodeling, and lipoprotein metabolism in animals [39,40]. Hu et al. [24] observed that, after reducing the fat contents of the feed, the lipase activity in the pancreas of 42-day-old meat ducks was significantly lower than that of the positive control group, suggesting that ducks may have the ability to regulate lipase activity based on the level of fat in their diet.

Many authors have evaluated the effects of emulsifier supplementation on serum lipid profiles in poultry, and the results have been inconsistent. Huang et al. [41] reported that soy lecithin supplementation resulted in a lower serum TC concentration in broilers on Day 42, while the concentrations of HDL-C and TG in the serum increased. In our study, serum HDL-C, LDL-C, and VLDL-C concentrations were not influenced by fat preE; however, fat preE tended to decrease the level of TC in the serum, suggesting that fat preE can affect serum lipid metabolism and improve the transport rate of serum TC in meat ducks. Different results were observed among these studies regarding lipid metabolism, which may be explained by the type of emulsifier used and the age of the poultry. Lysophospholipid supplementation decreased the TG, TC, and LDL-C concentrations in the serum of the ducks on Day 14, but these decreased biochemical indices were not observed on Day 28 [42]. Hu et al. [24] reported that adding complex emulsifiers (glycerine monostearate and polyoxyethylene sorbitol mono fatty acid ester) decreased the TG level in the serum on Day 42 but did not influence the TC level. Nevertheless, Upadhaya et al. [4] found that supplementation with different levels of 1,3-diacylglycerol had no effect on the TG concentration in broilers on Day 35.

Moreover, ALT and AST mainly exist in the cytoplasm of liver cells and are released into the blood when the liver is damaged, making them important indicators of liver function [6]. Interestingly, in this study, we found that the activity of AST in the serum was decreased by fat preE, suggesting that fat preE may improve the liver health of meat ducks. Consistently, the ducks fed diets with pre-emulsified fat were characterized by a lower liver index than the ducks receiving basal diets, which was in agreement with Ge et al. [43], who reported that the diets formulated with bile acids (BAs) markedly decreased the liver index of broilers on Days 21 and 42 by facilitating the transport of fat and alleviating hepatic fat deposition.

Finally, dietary supplemental lipids, in addition to supplying energy, can provide essential fatty acids and fat-soluble vitamins, reduce the pulverulence of pellets, and enhance the palatability of diets [44]. Haetinger et al. [5] demonstrated that supplementation with soybean oil sources did not influence the growth performance of broiler chickens from Days 0 to 42. Hulan et al. [45] also reported that the broilers fed diets supplemented with a combination of poultry fat and other animal fat (beef tallow or pork lard) exhibited similar growth performance when compared with those fed diets supplemented with a single fat source. This result was similar to ours, according to which the addition of different poultry fat sources had no significant effect on the growth performance of meat ducks. It is generally accepted that the FA composition of the fat source is a significant factor for the ATTD of fat or oils for poultry [18,46,47,48]. In fact, there were differences in the compositions of FAs between the duck fat and mixed poultry fat used in our present study. Similarly, Austic and Nesheim [49] reported that duck fat was rich in monounsaturated FAs, the levels of which were second only to olive oil and sunflower oil, but duck fat is short of long-chain polyunsaturated FAs when compared with broiler fat. In our study, we also found that the utilization of dietary EE was significantly higher and the serum TG concentration tended to decrease in the ducks fed the diets containing duck fat than for those containing mixed fat; thus, we assume that duck fat, as autologous fat, may provide more balanced FA make-up for ducks, thereby improving the utilization of dietary fat and the transport rate of serum TGs.

## 5. Conclusions

In conclusion, these results suggested that fat preE could improve the growth performance of ducks, which was related to nutrient utilization, digestive enzyme activities, serum lipid metabolism, and liver health. Notably, oil or fat preE is a better method to use to improve the utilization of feed oil or fat in poultry diets. Additionally, duck fat is more bioavailable for ducks based on dietary EE utilization and serum TG concentration.

## Figures and Tables

**Figure 1 animals-12-02729-f001:**
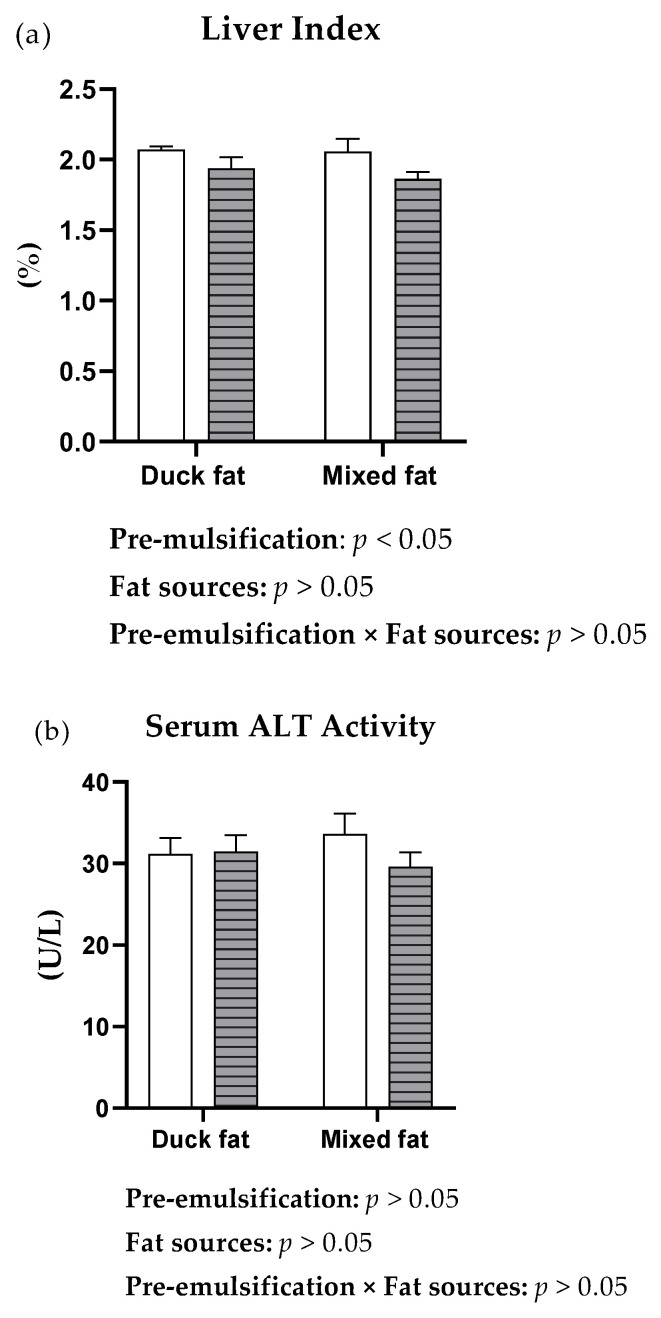
Effects of fat sources and fat pre-emulsification on the liver index (**a**), and the activity of ALT (**b**) as well as AST (**c**) in serum of 34-day-old meat ducks.

**Table 1 animals-12-02729-t001:** Composition and nutrient contents of the experimental diets (%, air dry basis) ^1–3^.

	Duck Fat		Mixed Fat	
Variables	TD	DP	TM	MP
Ingredients				
Corn	28.20	28.20	28.20	28.20
Wheat	24.10	24.10	24.10	24.10
Soybean meal	20.94	20.94	20.94	20.94
Wheat bran	5.10	5.10	5.10	5.10
Rice bran meal	12.00	12.00	12.00	12.00
Duck fat/Mixed fat	5.40	5.40	5.40	5.40
Bentonite	0.50	0.40	0.50	0.40
Emulsifier	—	0.10	—	0.10
Dicalcium phosphate	1.47	1.47	1.47	1.47
Calcium carbonate	1.13	1.13	1.13	1.13
*L*-Lysine. HCl (98.5%)	0.05	0.05	0.05	0.05
*DL*-Methionine (99%)	0.14	0.14	0.14	0.14
Sodium chloride	0.30	0.30	0.30	0.30
Choline chloride (50%)	0.15	0.15	0.15	0.15
Vitamin premix	0.03	0.03	0.03	0.03
Mineral premix	0.50	0.50	0.50	0.50
Total	100.00	100.00	100.00	100.00
Calculated nutrients, %				
ME, MJ/kg	12.14	12.14	12.14	12.14
Crude protein	17.50	17.50	17.50	17.50
Calcium	0.86	0.86	0.86	0.86
Ether extract	7.53	7.53	7.53	7.53
Total phosphorus	0.80	0.79	0.80	0.79
Available phosphorus	0.41	0.41	0.41	0.41
Total lysine	0.86	0.86	0.86	0.86
Total methionine	0.40	0.40	0.40	0.40
Total threonine	0.62	0.62	0.62	0.62
Total tryptophan	0.20	0.20	0.20	0.20
Analyzed nutrient levels, %				
Crude protein	17.05	17.06	17.03	17.08
Ether extract	6.79	6.83	6.71	6.83
Calcium	1.01	0.96	0.97	0.98
Total phosphorus	0.78	0.73	0.74	0.70

^1^ TD = diet supplemented with duck fat; DP = TD diet with fat pre-emulsification; TM = diet supplemented with mixed fat; MP = TM diet with fat pre-emulsification. ^2^ Vitamin premix provides the following per kg of final diet: vitamin A 8,000 IU; vitamin D_3_ 2,000 IU; vitamin E 5 mg; vitamin K_2_ 1 mg; vitamin B_1_ 0.6 mg; vitamin B_2_ 4.8 mg; vitamin B_6_ 1.8 mg; vitamin B_12_ 0.009 mg; niacin 10.5 mg; DL-calcium pantothenate 7.5 mg; folic acid 0.15 mg; ^3^ mineral premix provides the following per kg of final diet: Fe (FeSO_4_·H_2_O) 80 mg; Cu (CuSO_4_·5H_2_O) 8 mg; Mn (MnSO_4_·H_2_O) 70 mg; Zn (ZnSO_4_·H_2_O) 90 mg; I (KI) 0.4 mg; Se (Na_2_SeO_3_) 0.3 mg.

**Table 2 animals-12-02729-t002:** Fatty acid profile of duck fat and mixed fat (%) ^1,2^.

Item	Duck Fat	Mixed Fat
Content		
Lauric acid (C12:0)	0.20	0.10
Myristic acid (C14:0)	0.82	0.68
Palmitic acid (C16:0)	22.37	22.34
Palmitoleic acid (C16:1)	2.28	3.06
Margaric acid (C17:0)	0.39	0.34
Stearic acid (C18:0)	6.78	6.22
Elaidic acid (C18:1n9t)	0.29	0.24
Oleic acid (C18:1n9c)	42.36	41.46
Linoleic acid (C18:2n6)	21.49	22.97
γ-linolenic acid (C18:3n6)	0.11	0.14
α-linolenic acid (C18:3n3)	1.37	1.33
Arachidic acid (C20:0)	0.14	0.11
cis-11, 14-Eicosadienoic acid (C20:2)	0.39	0.25
cis-8, 11, 14-Eicosadienoic acid (C20:3n6)	0.16	0.11
Behenic acid (C22:0)	0.12	0.08
Saturated fatty acid (SFA)	31.11	30.09
Monounsaturated fatty acid (MUFA)	45.23	45.02
Polyunsaturated fatty acid (PUFA)	23.65	24.89
Unsaturated fatty acid to SFA (U:S ratio)	2.21	2.32

^1^ Number of carbon atoms and double bonds designated to the left and right of colon, respectively; ^2^ Mixed fat: chicken fat and duck fat in a 1:1 ratio.

**Table 3 animals-12-02729-t003:** Effects of fat sources and pre-emulsification on growth performance of ducks ^1,2^.

Fat Sources	Fat Pre-Emulsification	10 d BW/g	34 d BW/g	11–34 d BWG/g	11–34 d F: G	11–34 d FI/g
Duck fat	−	414	2181	1769	2.05	3614
	+	402	2268	1867	1.95	3670
Mixed fat	−	410	2196	1788	2.02	3577
	+	411	2238	1827	1.99	3634
	SEM	4.87	19.05	19.34	0.03	52.74
Main effects						
Duck fat		408	2228	1821	2.00	3642
Mixed fat		411	2218	1809	2.00	3605
	−	412	2189 ^b^	1779 ^b^	2.03 ^a^	3595
	+	407	2253 ^a^	1847 ^a^	1.97 ^b^	3652
	*p*-Value					
Fat sources		0.541	0.687	0.590	0.863	0.494
Pre-emulsification		0.257	0.002	0.002	0.021	0.293
Fat sources × Pre-emulsification		0.170	0.259	0.145	0.183	0.988

^a–b^ Values within a column with no common superscripts differ significantly (*p* < 0.05); ^1^ each value represents the mean value of 8 replicates/treatment (n = 8); ^2^ BW: body weight; BWG: body weight gain; FI: average daily feed intake; F/G: feed-intake-to-weight-gain ratio.

**Table 4 animals-12-02729-t004:** Effects of fat sources and pre-emulsification on serum biochemical index of ducks at 34 days of age ^1,2^.

Fat Sources	Fat Pre-Emulsification	TBA	TC	TG	HDL-C	LDL-C	VLDL-C
		(µmol/L)	(mmol/L)	(mmol/L)	(mmol/L)	(mmol/L)	(mmol/L)
Duck fat	−	31.18	4.99 ^a^	0.85	2.68	1.61	0.47
	+	19.03	4.25 ^b^	0.79	2.37	1.44	0.44
Mixed fat	−	20.10	4.27 ^b^	1.05	2.19	1.34	0.64
	+	21.74	4.25 ^b^	0.97	2.44	1.51	0.38
	SEM	4.70	0.19	0.11	0.15	0.13	0.09
Main effects							
Duck fat		25.11	4.59 ^a^	0.82	2.52	1.53	0.46
Mixed fat		20.87	4.26 ^b^	1.01	2.31	1.41	0.53
	−	24.85	4.55 ^a^	0.96	2.42	1.46	0.57
	+	20.49	4.25 ^b^	0.89	2.40	1.47	0.41
*p*-Value							
Fat sources		0.383	0.046	0.085	0.163	0.416	0.539
Pre-emulsification		0.276	0.037	0.490	0.826	0.983	0.110
Fat sources × Pre-emulsification		0.157	0.047	0.936	0.065	0.198	0.201

^a–b^ Values within a column with no common superscripts differ significantly (*p* < 0.05); ^1^ each value represents the mean value of 8 replicates/treatment (n = 8); ^2^ TBA: total bile acid; TC: total cholesterol; TG: triglyceride; HDL-C: high-density lipoprotein; LDL-C: low-density lipoprotein; VLDL-C: very low-density lipoprotein.

**Table 5 animals-12-02729-t005:** Effects of fat sources and pre-emulsification on the activity of trypsin and lipase in jejunum as well as in duodenum of ducks at 34 days of age ^1^.

Fat Sources	Fat Pre-Emulsification	Jejunum		Duodenum	
		Trypsin (U/mg prot)	Lipase (U/mg prot)	Trypsin (U/mg prot)	Lipase (U/mg prot)
Duck fat	−	72,254	71.37	57,887	52.72
	+	58,857	39.87	30,027	24.74
Mixed fat	−	81,381	63.32	58,902	57.6
	+	81,291	76.88	55,062	50.51
	SEM	6932	12.86	6209	9.16
Main effects					
Duck fat		65,555 ^b^	55.62	44,885 ^b^	37.79
Mixed fat		81,336 ^a^	70.10	56,982 ^a^	54.29
	−	76,818	67.35	58,394 ^a^	55.32
	+	70,074	58.37	43,379 ^b^	36.76
*p*-Value					
Fat sources		0.031	0.270	0.046	0.106
Pre-emulsification		0.339	0.491	0.017	0.067
Fat sources × Pre-emulsification		0.345	0.091	0.064	0.265

^a–b^ Values within a column with no common superscripts differ significantly (*p* < 0.05); ^1^ each value represents the mean value of 8 replicates/treatment (n = 8).

**Table 6 animals-12-02729-t006:** Effects of fat sources and pre-emulsification on nutrients and energy utilization of ducks ^1,2^.

Fat Sources	Fat Pre-Emulsification	Dry Matter (%)	EE (%)	Energy (%)	AME (kcal/kg)	Crude Protein (%)	TP (%)	Ca (%)
Duck fat	−	70.20	86.79	74.56	2918	60.49	33.70	37.34
	+	71.72	89.50	75.89	2968	64.90	40.93	38.56
Mixed fat	−	69.01	82.20	73.44	2875	61.76	28.28	41.14
	+	71.42	87.45	75.59	2960	66.87	38.70	40.79
	SEM	0.79	1.43	0.67	26.15	2.84	2.74	4.30
Main effects								
Duck fat		71.06	88.33 ^a^	75.27	2945	62.84	37.56	37.99
Mixed fat		70.21	85.00 ^b^	74.51	2917	64.32	33.84	40.97
	−	69.52 ^b^	84.31 ^b^	73.92 ^b^	2893 ^b^	61.17	30.99 ^b^	39.37
	+	71.57 ^a^	88.47 ^a^	75.73 ^a^	2964 ^a^	65.89	39.82 ^a^	39.68
*p*-Value								
Fat sources		0.360	0.029	0.298	0.333	0.573	0.175	0.489
Pre-emulsification		0.020	0.010	0.015	0.016	0.105	0.003	0.920
Fat sources × Pre-emulsification		0.581	0.383	0.541	0.504	0.902	0.567	0.856

^a–b^ Values within a column with no common superscripts differ significantly (*p* < 0.05); ^1^ each value represents the mean value of 8 replicates/treatment (n = 8); ^2^ EE: ether extract; AME: apparent metabolizable energy; TP: total phosphorus; Ca: calcium.

**Table 7 animals-12-02729-t007:** Effects of fat sources and pre-emulsification on standardized ileal digestibility of nonessential amino acids of ducks at 40 days of age (%) ^1,2^.

Fat Sources	Fat Pre-Emulsification	Asp	Ser	Glu	Gly	Ala	Cys	Tyr	Pro	Total NEAA
Duck fat	−	59.06	65.36	76.64	55.58	55.80	87.95	59.75	74.70	67.91
	+	58.10	69.60	74.15	52.72	55.35	86.42	60.61	72.46	66.86
Mixed fat	−	62.98	66.22	77.37	56.35	57.54	79.78	68.30	74.33	69.60
	+	63.91	68.27	78.31	58.51	60.56	84.87	67.73	72.48	70.92
	SEM	1.96	1.78	1.19	2.33	2.25	2.22	2.47	1.40	1.60
Main effects										
Duck fat		58.61^b^	67.34	75.47 ^b^	54.25	55.59	87.19 ^a^	60.15 ^b^	73.58	67.41
Mixed fat		63.45^a^	67.25	77.84 ^a^	57.50	59.15	82.32 ^b^	68.01 ^a^	73.41	70.26
	−	60.89	65.76	76.98	55.94	56.61	83.86	63.74	74.51	68.70
	+	61.01	68.93	76.23	55.81	58.13	85.64	64.17	72.47	68.89
*p*-Value										
Fat sources		0.020	0.898	0.050	0.173	0.136	0.038	0.004	0.903	0.083
Pre-emulsification		0.996	0.090	0.524	0.882	0.571	0.430	0.955	0.157	0.933
Fat sources × Pre-emulsification		0.634	0.543	0.161	0.292	0.449	0.150	0.775	0.889	0.465

^a–b^ Values within a column with no common superscripts differ significantly (*p* < 0.05); ^1^ each value represents the mean value of 8 replicates/treatment (n = 8); ^2^ Asp: aspartic acid; Ser: serine; Glu: glutamic acid; Gly: glycine; Ala: alanine; Cys: cysteine; Tyr: tyrosine; Pro: proline; total NEAA: total nonessential AA.

**Table 8 animals-12-02729-t008:** Effects of fat sources and pre-emulsification on standardized ileal digestibility of essential amino acids of ducks at 40 days of age (%) ^1,2^.

Fat Sources	Fat Pre-Emulsification	Thr	Val	Met	Ile	Leu	Phe	Lys	His	Arg	Total EAA	Total AA
Duck fat	−	50.85	57.26	59.38	58.56	59.88	65.13	52.89	63.01	68.50	59.88	64.57
	+	55.90	54.45	60.44	56.24	59.53	64.23	51.34	61.03	69.47	59.36	63.69
Mixed fat	−	54.52	62.09	55.69	64.30	64.23	69.09	59.63	66.69	70.03	63.79	67.15
	+	55.70	61.59	59.14	63.35	63.99	69.85	60.66	67.73	71.90	65.31	68.56
	SEM	2.54	2.25	3.13	2.47	2.17	1.73	2.27	1.90	1.87	2.06	1.78
Main effects												
Duck fat		53.20	55.95^b^	59.88	57.48 ^b^	59.71	64.71 ^b^	52.23 ^b^	62.08 ^b^	68.95	59.64 ^b^	64.16 ^b^
Mixed fat		55.15	61.82^a^	57.53	63.83 ^a^	64.11	69.47 ^a^	60.18 ^a^	67.21 ^a^	70.97	64.55 ^a^	67.86 ^a^
	−	52.56	59.51	57.66	61.24	61.91	66.97	56.03	64.73	69.21	61.70	65.77
	+	55.79	58.26	59.75	59.80	61.76	67.04	56.67	64.38	70.69	62.34	66.13
*p*-Value												
Fat sources		0.500	0.013	0.432	0.016	0.053	0.011	0.002	0.011	0.299	0.025	0.047
Pre-emulsification		0.232	0.468	0.478	0.516	0.893	0.969	0.910	0.807	0.452	0.810	0.882
Fat sources × Pre-emulsification		0.452	0.612	0.706	0.785	0.982	0.637	0.575	0.435	0.811	0.625	0.528

^a–b^ Values within a column with no common superscripts differ significantly (*p* < 0.05); ^1^ each value represents the mean value of 8 replicates/treatment (n = 8); ^2^ Thr: threonine; Val: valine; Met: methionine; Ile: isoleucine; Leu: leucine; Phe: phenylalanine; Lys: lysine; His: histidine; Arg: arginine; total EAA: total essential AA.

## Data Availability

Data are available upon request from the corresponding authors.

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
