# Peer review of "Effects of Fat Pre-Emulsification on the Growth Performance, Serum Biochemical Index, Digestive Enzyme Activities, Nutrient Utilization, and Standardized Ileal Digestibility of Amino Acids in Pekin Ducks Fed Diets with Different Fat Sources"

_animals, 2022, doi:10.3390/ani12202729_

Round 1
Reviewer 1 Report
The article provides useful information about effects of fat pre-emulsification and their impact on growth performance and nutrient utilisation in ducks. There are however some critical points to be addressed:
Introduction
The introduction did not clearly present the hypothesis of this study.
Material and methods
It seems that you don’t have a control group, where a ducks should have diets with soybean oil. I think it’s major flaw.
In table 2 the abbreviation of poluunsaturetad fatty acid should be “PUFA”
“The ducks from which blood was collected were euthanized by exsanguination. 149 Liver weights were obtained and calculated based on live BW. After that, the digesta 150 from the duodenum and jejunum were collected and stored at -80 °C for enzyme activity 151 analysis.” are the same birds you used for digesta collection ? Is there no interference on this procedure of collection ?
L.162 How looks analyze of DM, EE, CP,Ca, TP? It is not enough to write references. Which appliance has been used?
References:
According to the Journal Style is necessary to write abbreviated Journal Name in references, for example 223
Author Response
We do really appreciate the comments from you. Thank you for your hard work! We have provided a point-by-point response to your comments. Please see the attachment.

Reviewer 2 Report
Overall, the manuscript addresses the effects of fat pre-emulsification on growth perfomance, serum biochemical index, digestive enzyme activities, nutrient utilization, and SID of amino acids in Pekin Ducks. It is well written.
However, minor revisions are required for this manuscript.
Line 20: add 'of' for 'With increasing of oil prices'
Line 26: delete 'first'
Line 28: delete the first abbreviation 'SID'
Lines 36 and throughout the manuscript, please use only 2 decimals for P-values.
Line 54: the requirements in metabolizable energy
Line 68: use 'strain' instead of 'line' and delete 'the' before poultry
Line 98: delete 'first'
Line 99: delete 'based'
Line 108: use 409+/-27 g
Line 111: I recommend to use this sentence 'of 8 replicate cages with 10 ducks per cage'.
Table 1: I suggest to modify the names of treatments and clarify what behind fat in feed composition in order to make the treatments more understandable. BD1 and BD2 not 'BD' for both as well as for BP.
For synthetic amino acids such as DL-methionine and L-lysine HCl, , please add the % of methionine and lysine, respectively.
Use 'Calculated nutrients' without 'levels'
Add calculated 'Total phosphorus' values
Precise for amino acid values (lysine, methionine, threonine and tryptophan) if they are total or digestible.
Use capital letter for 'total phosphorus'
Line 178: Describe in more detail the calculation of standaridized ileal digestibility of amino acids.
Table 3: Add the intial BW of ducks.
Lines 200-201: check the figure 1 and add superscript letters for liver index.
Table 6: 'kcal' instead of 'Kcal'
Author Response
We do really appreciate the suggestions from you. Thank you for your hard work! We have revised the manuscript according to your suggestions one by one. Please see the attachment.

Round 2
Reviewer 1 Report
The manuscript can be accept in present form.